

# Hermit crab response to a visual threat is sensitive to looming cues

Talya Shragai[1], Xiaoge Ping[2], Cameron Arakaki[3], Dennis Garlick[3], Daniel T. Blumstein[1] and Aaron P. Blaisdell[3]

[1] Department of Ecology and Evolutionary Biology, University of California, Los Angeles, CA, United States of America

[2] Key Laboratory of Animal Ecology and Conservation Biology, Institute of Zoology, Chinese Academy of Sciences, Beijing, China

[3] Department of Psychology, University of California, Los Angeles, CA, United States of America

## ABSTRACT

Prior work in our lab has shown that an expanding image on a computer screen elicits a hiding response in the Caribbean terrestrial hermit crab *(Coenobita clypeatus)*. We conducted two experiments to identify what properties of the expanding stimulus contribute to its effectiveness as a visual threat. First we found that an expanding geometric star evoked a strong hiding response while a contracting or full-sized stationary star did not. A second experiment revealed that the more quickly the stimulus expanded the shorter the latency to hide. These findings suggest that the anti-predator response to looming stimulus relies heavily on visual cues relating to the manner of approach. The simulated visual threat on a computer screen captures key features of a real looming object that elicits hiding behavior in crabs in the wild.

## INTRODUCTION

Corresponding author
Aaron P. Blaisdell,
blaisdell@psych.ucla.edu

Many species rely on vision for threat detection. The ability to appropriately respond to an approaching predator requires the ability to detect the visual threat and estimate its imminence (*Fanselow & Lester, 1988*). In a natural context, rapidly approaching visual stimuli elicit the looming response (*Chan et al., 2010a*). It can be challenging to experimentally manipulate individual features of the looming stimulus to determine their effectiveness. By using a looming image on a computer screen, it is possible to simulate an approaching predator in controlled laboratory conditions and measure the nature and timing of responses. Using such methods, many species have been shown to flee or hide from looming images (e.g., *Chan et al., 2010b*; *Fotowat & Gabbiani, 2011*; *Preuss et al., 2006*; *Rind & Simmons, 1999*; *Sun & Frost, 1998*; *Yilmaz & Meister, 2013*).

But not all looming images are associated with an increased predation risk. Thus, other studies have manipulated the image content to determine whether a variety of images are similarly evocative. For instance, *Curio (1975)* manipulated the structure of a looming predator image to show that pied flycatchers *(Ficedula hypoleuca)* used a complex combination of visual cues to discriminate both between predators and non-predators, and between similar types of predators. Similar discriminative abilities have been reported from

observational field experiments. For instance, *Walters (1990)* showed that three species of lapwings (Charadriidae: *Vanellus* spp.) also use visual cues to identify predators while *Seyfarth & Cheney (1990)* demonstrated that vervet monkeys (*Chlorocebus pygerythrus*) emitted different alarm calls in response to the sight of different predators (see *Yilmaz & Meister, 2013*, for similar results in mice).

Species are also sensitive to the rate at which an object expands because rate contains potential information about the time to impact. Studies of crayfish (*Procambarus clarkii*, *Glantz, 1974*), fish (*Carassius auratus*, *Preuss et al., 2006*), and fiddler crabs (*Neohelice granulate*, *Hemmi & Pfeil, 2010*; *Oliva & Tomsic, 2012*) have measured response differentiation when the speed or size of the looming image was changed, showing that species are sensitive to alter the timing of their response based on calculations of the timing of collision.

Caribbean terrestrial hermit crabs *(Coenobita clypeatus)* are an ideal invertebrate system to test looming image discrimination and processing. Prior work in our lab demonstrated that hermit crabs have a binary escape response; they hide (or fail to hide) in their shells when presented with a looming image of a predator or geometric shape presented on a computer screen (*Chan et al., 2010b*; *Ping et al., 2015*; *Ryan et al., 2012*; *Stahlman et al., 2011*; *Watanabe et al., 2012*).

Using procedures developed in our lab, we designed two experiments to manipulate specific aspects of the image and then measured responses by recording the number of trials to habituation, the latency to hide, and the latency to emerge from their shells. The goal of the experiments was to determine which factors contributed to the effectiveness of the looming visual stimuli used in our prior research to elicit antipredator hiding responses. The first experiment compared the typical looming stimulus produced by the gradual expansion of the image with two other conditions. In one condition, the stimulus was presented suddenly at full size, and in the other the stimulus began at full size and contracted to a single pixel. The second experiment tested how changing the rate of image expansion affected response times.

## GENERAL METHODS

The subjects were 57 hermit crabs acquired from a pet supply company (California Zoological Supply). Experimentally-naïve crabs were used for each experiment. The hermit crabs' shells aperture length ranged from 3 to 4 cm across. Crabs were housed in UCLA's Comparative Cognition Laboratory (http://pigeonrat.psych.ucla.edu) in groups of one to six. Each had its claw numbered with non-toxic OPI nail polish for identification. Subjects were housed in 50 × 25 × 25 cm clear plastic bins. Bins were lined with coconut fiber substrate (Zoo Med Eco Earth, San Luis Obispo, CA, USA) and each contained two ceramic water dishes, a paper plate under which the crabs could comfortably rest, a wet sponge to maintain humidity, and a plastic food cup. Crabs always had access to distilled drinking water in one dish and a 1% salt solution in the other, and were given Tetrafauna Hermit Crab meal pellets daily (Blacksburg, VA, USA). The coconut substrate was checked daily for fungus and changed three times a week or more often as necessary to prevent infection.

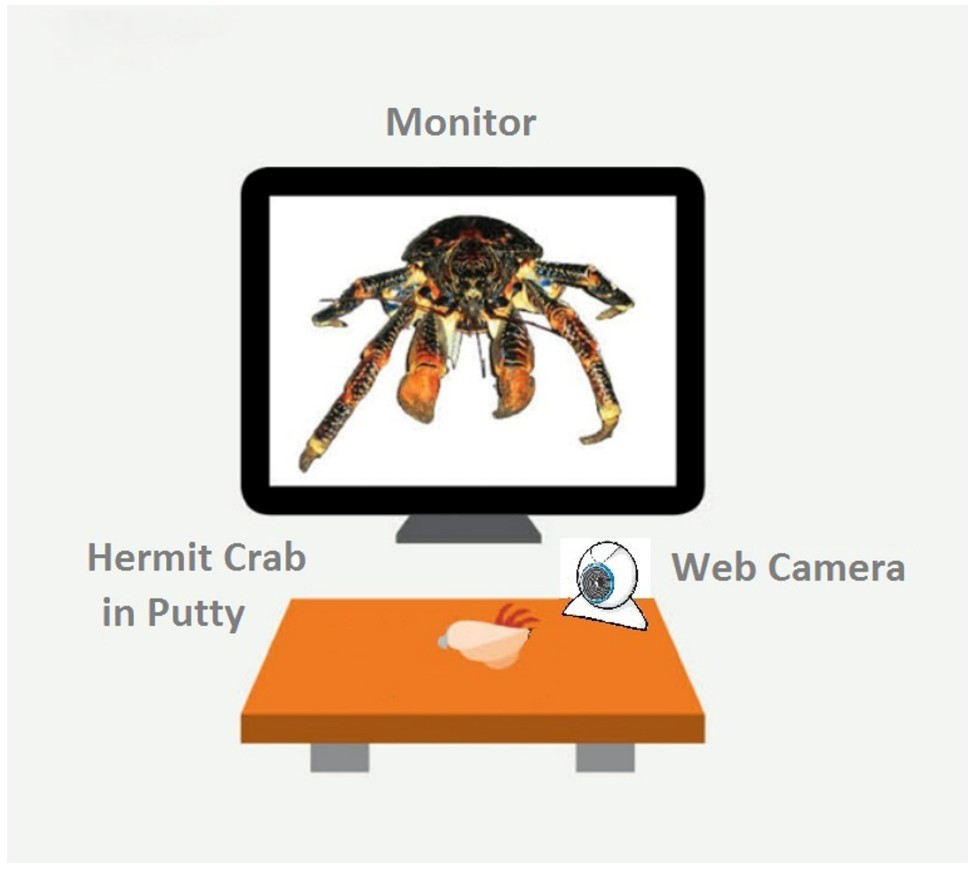

**Figure 1  A schematic of our experimental set-up.** The monitor displays the fully enlarges image of the coconut crab stimulus. Figure not drawn to scale.

The dishes containing water were cleaned and replenished daily, and the coconut substrate was replaced on Mondays, Wednesdays, and Fridays. Each bin was mostly covered with a clear acrylic sheet (0.5 cm thick) to maintain humidity, with a small opening to allow air circulation. The room contained a Vicks humidifier (Proctor and Gamble, Cincinnati, OH, USA) to maintain the atmosphere at 50–75% humidity in the bins. Two 9.5 Watt heat lamps were used to provide warmth for the bins, maintaining a room temperature of 25 °C. A timer provided a 14:10 h light: dark cycle, with lights turning on at 8:00 AM.

All experimental trials were performed in a test arena located on a desktop (Fig. 1). We constructed a wooden, matt black arena 18.2 cm W × 30.5 cm L, ×14.5 cm H). A 15-inch LCD monitor (Dell 1704FPVt; Dell, Round Rock, TX, USA) with a screen resolution of 1,024 × 768 pixels was used to display visual stimuli. Subjects were secured in a crab-holder using Fun Tak[TM] blue and white putty (Henkel Removable Poster Putty[TM], model 1084014; Henkel, Duseldorf, Germany) attached to the crab's shell. The holder was positioned in front of and 13.3 cm away from the LCD screen. Subjects were oriented so that the aperture of their shell was angled up and perpendicular in relation to the ground when secured.

We used a custom program developed in-house to automatically recorded latency to hide and re-emerge (*Ryan et al., 2012*). A web camera (model C250; Logitech, Lausanne, Switzerland) was placed 5 cm in front of the clamp, below the LCD monitor. The program detected activity in a 300 × 150 pixel region in the camera's visual field. At the beginning of a session, the subject was placed on its back into the putty and clipped into the apparatus. The square-shaped visual detection field was positioned so that the bottom 1/3rd contained the crab's shell and the top 2/3rds was empty to allow detection space for when the crab emerged. We then set the number of "hiding pixels" (i.e., the number of pixels the detector registered when the crab was hiding), and the program computed a value to set a variable called "ExtraPixelsOut" to determine when the crab had emerged from its shell. The value of "ExtraPixelsOut" was calibrated for each individual crab at the beginning of each session to account for each individual crab's size. Furthermore, to account for variation in individual size, we calculated a variable labeled "ClawSize" for each crab. To calculate "ClawSize" we set the baseline for a hiding crab at zero pixels, recorded the minimal number of pixels once the crab emerged, subtracted 1,000 from this number, and divided it by 0.5. After this calibration was complete, a crab was scored as hiding until this additional number of pixels was detected in the recording square on the monitor. We found this method to reliably discriminate the state of the crab being emerged from its shell from the state of it hiding within its shell (*Ryan et al., 2012*).

A variable labeled "ExtraPixelsIn" was computed to ensure the program accurately registered when the crab returned to hiding. After emerging from its shell, a hiding response did not always result in the crab returning as far into its shell as when the initial "hiding pixels" baseline was set at the beginning of the session. Thus, to increase sensitivity to detecting hiding responses, "ExtraPixelsIn" was set to 1/6th the hiding pixels. Once the crab emerged, the program did not consider the crab as being "hiding" until the number of pixels detected fell below this number of additional pixels (*Ryan et al., 2012*).

The criterion for number of trials to habituation was defined as two consecutive trials with no registered hide response. Both experiments were video recorded and the results of the automatic detector were verified on the video and hand scored when necessary to include cases where the automatic detector failed to detect response.

If a subject failed to emerge for 20 min at the beginning of a session it was returned to its home cage and was tried again after all the other crabs had completed their session for the day. If a hermit crab still failed to emerge from its shell, it was not included in the experiment.

## EXPERIMENT 1

In this experiment we compared the effectiveness of an expanding stimulus, a contracting stimulus, and a full-sized stationary stimulus on eliciting the hide response.

### Methods

*Subjects.* We tested 30 experimentally-naïve crabs. 10 crabs were tested on day 1, another 10 were tested on day 2, and the remaining 10 crabs were tested on day 3 of the experiment. Each day, about 1/3rd of the crabs were tested in each of the three conditions: Expanding,

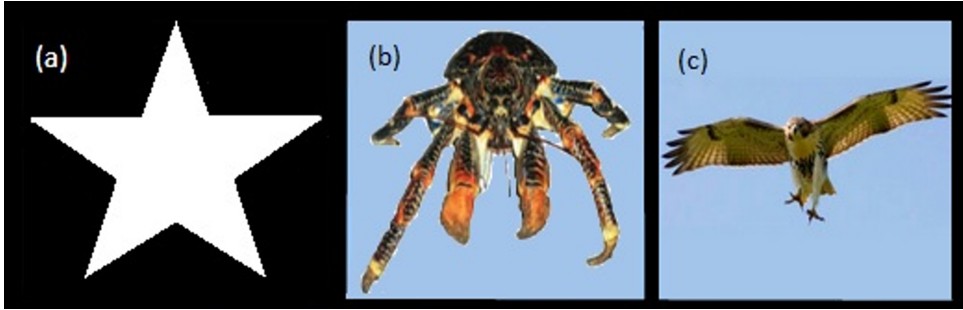

**Figure 2** **Visual stimuli.** Star image for Experiment 1 (A), Hawk and Coconut Crab images for Experiment 2 (B, C).

Contracting, and Static. Test order was counterbalanced using a Latin square design to account for confounding effects of time of day and day of the week.

*Stimuli.* The stimulus consisted of a white star shape, 1,024 × 1,024 pixels, on a black background (Fig. 2A).

*Procedure.* A session started with the crab being placed in the apparatus as described in the General Methods. Each trial began as soon as the crab was automatically detected as being out of its shell for 30 consecutive seconds. If the program detected that the crab hid again during this interval, the pre-trial timer would reset and not restart until the crab was detected as being "out" again. Once the crab had been detected as "out" for 30 consecutive seconds, the trial began with a 30-s delay followed by the presentation of the visual stimulus. Figure 3 provides a schematic for the time course of each of the three types of presentation of the stimulus. The Expanding image started as a single pixel at the top of the screen and enlarged at an exponentially increasing rate such that it simulated the approach of the stimulus at a constant rate. From the start presentation, the stimulus reached its full width of approximately 900 pixels in 17 s. Once the stimulus reached its largest size, it filled the lower part of the screen and remained on the screen for an additional 5 s. For Static presentations, the stimulus was presented at full size at the center of the screen for 22 s. For Contracting presentations, the stimulus was initially presented at its full size and contracted at an exponentially decreasing rate to simulate the image receding at a constant rate. The contracting presentation took 17 s to go from full size to a single pixel at the top of the screen, after which the single pixel remained on the screen for an additional 5 s.

For all conditions, trials were repeated until either the visual stimulus had been presented 15 times, the session reached a time ceiling of 30 min, or until the crab had two consecutive trials on which it did not hide at all during the stimulus presentation (i.e., reached the habituation criterion).

## Results

Very few subjects in the Contracting or Static conditions hid to the stimulus presentation on any trial, precluding a group comparison of rates of habituation. Therefore, we instead analyzed proportion of crabs that hid on the first trial in each of the three conditions using

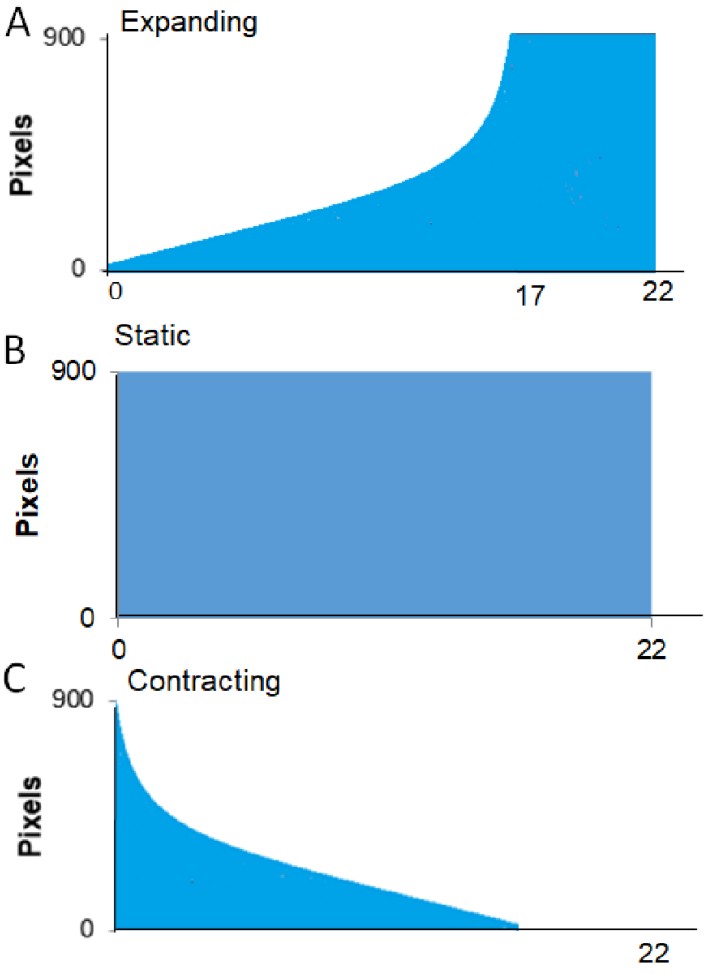

**Figure 3 Presentation formats for Experiment 1.** Schematic of the conditions, expanding, full-sized, and contracting stimulus for Experiment 1.

a two-tailed Freeman-Halton extension of Fisher's exact test. For all tests, we interpret two-tailed $p$-values <0.05 as significant.

Stimulus presentation had a large effect on hiding responses ($p = 0.023$; Fig. 4), with the Expanding stimulus eliciting the most reliable response. A larger proportion of crabs in the Expanding condition hid as compared to the Static or Contracting conditions (Fisher's exact test: $p = 0.023$). There was no difference in the proportion of crabs that had in the Static or Contracting conditions ($p = 1.000$). These results suggest that the expanding stimulus was perceived as much more threatening than either the static or contracting stimuli.

## EXPERIMENT 2

In this experiment we evaluated the effect of varying rate of stimulus expansion on probability and latency to hide. Rate of expansion correlates with the changes in the
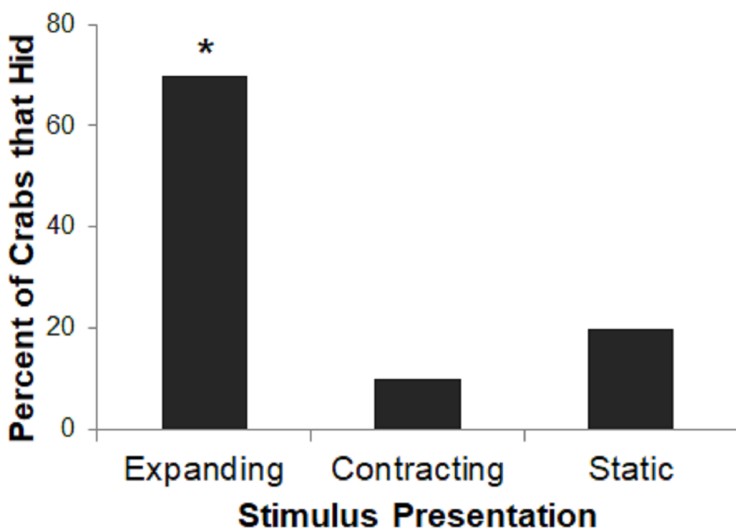

**Figure 4** **Percent of crabs that hid to the Expanding, Contracting, and Static conditions in Experiment 1.** Asterisk indicates significant difference from remaining groups, *p* < .05.

amount of stimulus that fills the visual field of the perceiver of a real looming object. Thus, to the degree that rate of approach of a looming object elicit differences in latency and probability of a hiding response, similar effects should be observed when rate of expansion of a two-dimensional stimulus on a computer screen is varied.

## Methods

*Subjects.* We tested 26 experimentally-naïve crabs randomly allocated to one of three groups (*n* = 8 for Group 8 s and 32 s, and *n* = 10 for Group 16 s). Subjects were maintained as in Experiment 1.

*Stimuli.* To assess the generality of the effect of expansion, two stimuli were used; an image of a red-tailed hawk (*Buteo jamaicensis*, 1,024 × 796 pixels at full size) and the image of a coconut crab (*Birgus latro*, 1,024 × 430 pixels at full size, Figs. 2B, 2C). All other aspects of the apparatus were as described for Experiment 1.

*Procedure.* Each subject received one session per day for two days. Each crab received one session with the hawk stimulus and one session with the coconut crab stimulus, with test order counterbalanced across subjects such that half of each group was shown the hawk on the first day and the coconut crab on the second day while the order was reversed for the remaining crabs. Groups differed in the rate of stimulus expansion from onset to reaching full size. The three expansion rates tested were 8 s (fast velocity), 16 s (medium velocity), and 32 s (slow velocity). Expansions occurred exponentially such that, in each condition, the stimulus appeared to approach at a constant rate throughout the presentation.

Each session began and trials were conducted as described for Experiment 1. Each trial consisted of the expansion from a single pixel to a full size image of either a coconut crab picture or a hawk picture. Trials with the same visual stimulus were repeated until either the visual stimulus had been presented 15 times, the session reached a time ceiling of 30

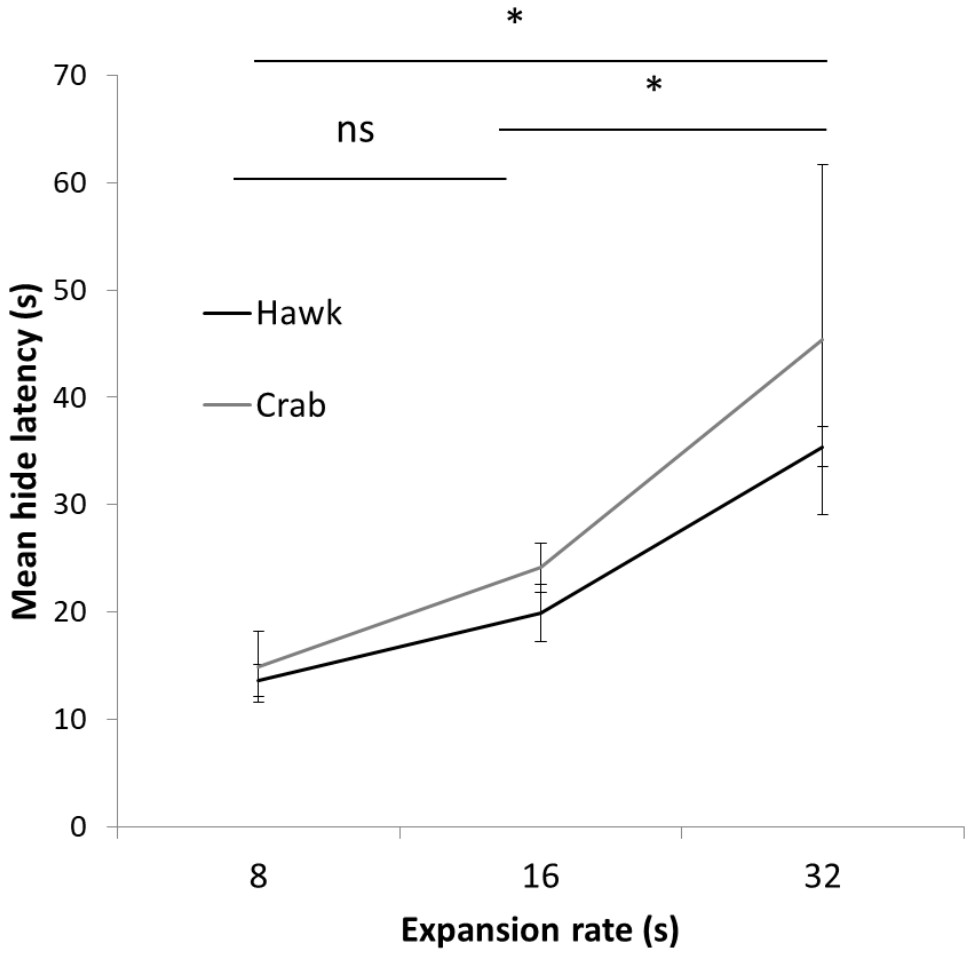

**Figure 5 Results Experiment 2.** Mean latency (s) to hide to the 8 s, 16 s, and 32 s presentation time stimuli using both a Hawk and Coconut Crab image in Experiment 2. Error bars depict standard errors of the means.

min, or until the crab had two consecutive trials on which it did not hide at all during the stimulus presentation (i.e., reached the habituation criterion).

## Results

A two-way mixed ANOVA conducted on latencies to hide on the first trial with Expansion Rate (8 s, 16 s, and 32 s) as a between-subject factor and Stimulus Type (Hawk and Coconut Crab) as a repeated measure revealed a main effect of Expansion Rate $F(1, 23) = 7.05$, $p < 0.01$, $\eta p^2 = 0.38$, but no effect of Stimulus Type, $F(1, 23) = 1.06$, $p = 0.31$, nor interaction between Expansion Rate and Stimulus Type, $F < 1.0$. Planned comparisons revealed hide latencies differed between the 8 s and 32 s presentation times $F(1, 23) = 13.24$, $p = 0.01$, and between the 16 s and 32 s presentation times $F(1, 23) = 7.29$, $p < 0.05$ (Fig. 5), but not between the 8 s and 16 s presentation times, $F(1, 23) = 1.29$, $p = .27$. A similar ANOVA conducted on latencies to emerge failed to reveal any main effects or interactions, $Fs(1, 22) < 1.58$, $ps > 0.23$ (latency to emerge data were missing from one subject in

|  | **Hawk** | | |
|---|---|---|---|
|  | **Expansion rate (s)** | | |
|  | **8** | **16** | **32** |
| **Habituated** | 37.5% | 60% | 50% |
| **Not habituated** | 62.5% | 40% | 50% |

**Figure 6** Percentage of subjects that reached the habituation criterion in each condition of Experiment 2.

Group 16 s, and thus $n = 23$ for this analysis). Because a large number of subjects failed to habituate, trials to habituation was not analyzed as a measure of reaction (Fig. 6). These results suggest that rate of expansion serves as a proxy for rate of approach of a looming stimulus, with faster rates eliciting shorter latencies to hide.

## DISCUSSION

An expanding image was much more effective than a static or contracting image at eliciting the anti-predatory hide response in hermit crabs. This supported the hypothesis that hermit crabs perceived the expanding image as threatening and probably as a looming object. Moreover, the faster the rate of expansion, the shorter the latency to hide, which is also consistent with how animals react to looming stimuli under natural conditions. Collectively, these experiments demonstrate that for Caribbean terrestrial hermit crabs, the visual system plays an essential role in identifying and responding to potential threats. While dynamic features of an expanding stimulus likely simulate the key aspects of a looming stimulus, it is difficult to determine whether crabs perceived the pictorial or representational aspects of the images. Many species show mixed evidence of perceiving photographs or pictures presented on a computer screen as representations of real 3D objects versus as only 2D colored patterns. For example, evidence is mixed as to whether pigeons (*Columba livia*) can perceive the pictorial or representational aspects of pictorial displays (*Fagot, 2000*). We've found evidence that the Caribbean hermit crab lacks color vision (*Ping et al., 2015*). Unpublished studies from our lab also fail to find evidence that hermit crabs can distinguish between pictorial content that is naturalistic (e.g., photographs of a hawk, coconut crab, sea gull), man-made real objects (e.g., a picture of a couch), and simple geometric shapes (e.g., a square, star, or oval). More research would be needed to definitively answer this question for the Caribbean hermit crab. Nevertheless, the fact that hermit crabs were particularly sensitive to an expanding stimulus suggests that some aspects of real, looming objects are perceived in the dynamic 2D stimuli in our study. Our results are consistent with much prior work exploring the neural basis of antipredator responses to looming stimuli in arthropods (*Hemmi, 2005*; *Oliva, Medan & Tomsic, 2007*; *Oliva & Tomsic, 2012*).

While we documented the visual detection abilities that these crabs possessed, previous experiments have shown the importance of the acoustic environment to the crab's ability to

respond to visual stimuli (*Chan et al., 2010b*; *Ryan et al., 2012*), indicating that hermit crabs use multisensory channels to evaluate their environment. This is consistent with research on other invertebrates such as fiddler crabs (*Uca vomeris*) that employ multiple detector systems to evaluate different aspects of a possible threat (*Hemmi, 2005*; *Hemmi & Pfeil, 2010*). Future studies would profitably examine how assessment works in a multi-modal paradigm. Finally, it is worth noting that this and prior published work from our lab demonstrates the value of going from field research into the lab and back again. Field research is valuable for its ability to gather questions about an organism's behaviors in its natural habitat. Laboratory research can then provide a more controlled setting within which to untangle distinct behavioral processes.

## CONCLUSION

Computer technology can provide a valuable tool for investigating the perceptual and cognitive processes that contribute to natural behavior observed in the field. Knowledge gathered from laboratory work can in turn inform new directions for field work, including tailoring improvements that protect animals from potentially deleterious anthropogenic effects.

## ACKNOWLEDGEMENTS

Special thanks to Hwee Cheei Lim for her constant support and encouragement.

### Funding

This study was supported by NSF Research Grant BCS-0843027 (to Aaron Blaisdell). Daniel Blumstein is supported by the NSF. The funders had no role in study design, data collection and analysis, decision to publish, or preparation of the manuscript.

### Grant Disclosures

The following grant information was disclosed by the authors:
NSF Research: BCS-0843027.

### Competing Interests

The authors declare there are no competing interests.

### Author Contributions

- Talya Shragai conceived and designed the experiments, performed the experiments, analyzed the data, wrote the paper, prepared figures and/or tables, reviewed drafts of the paper.
- Xiaoge Ping performed the experiments, contributed reagents/materials/analysis tools.
- Cameron Arakaki performed the experiments.
- Dennis Garlick contributed reagents/materials/analysis tools.

- Daniel T. Blumstein conceived and designed the experiments, wrote the paper, reviewed drafts of the paper.
- Aaron P. Blaisdell conceived and designed the experiments, contributed reagents/materials/analysis tools, wrote the paper, prepared figures and/or tables, reviewed drafts of the paper.

## Data Availability

The raw data has been provided in Supplemental Files.

## Supplemental Information

Supplemental information for this article can be found online at http://dx.doi.org/10.7717/peerj.4058#supplemental-information.

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
