# Peer review of "Hermit crab response to a visual threat is sensitive to looming cues"

_PeerJ, doi:10.7717/peerj.4058_

## Round 0.1 · original submission · Minor Revisions

· Academic Editor

Minor Revisions

I do agree with all suggestions from both reviewers, especially about the comparison between the star stimulus in exp 1 and the more naturalistic stimuli in exp 2. The proportion of crabs that hide in each condition in exp 2 would at least provide potential comparison.

I do have also a few additional remarks to take into consideration when resubmitting your manuscript:

1- page 7: 'Thus, we also counted instances of freezing and partial hiding as responses to visual threat' - How often these types of behaviours were observed?
2- page 8: The description of the subjects is not clear 'We tested 10 experimentally-naïve crabs on each of three consecutive days. Each crab was only tested on a single day, and the 30 crabs were randomly split into three groups of 10: Expanding, Contracting, and Static. Test order was counterbalanced' It is not clear what the 10 naive crabs did. It is not clear either why independent groups would have a test order.
3- results of exp 1 (fig 4): the results are analyzed as 'proportion of crabs that hid' - there were 15 trials - so what was the criterion for categorizing a crab has hidding ?
4- Exp 2, page 10: The number of subjects does not sum to 27 crabs (8+8+10)
5- Exp 2, results: How were the mean latencies calculated, as the number of trials were potentially not equivalent for all the crabs (because they were stopped if they have reached the habituation criterion).
6- Exp 2, figure 5: the errors bars are huge for 'crab - 32s expansion rate' condition - why? were there two population of animals?

Reviewer 1 ·

Basic reporting

Figures 1 and 2 don't contain any scale information. Authors should provide scale.

Figure 3: Authors suggest that the stimuli expanded and contracted exponentially to mimic a predator approach with constant speed (lines 160, 166). This is inconsistent with this figure where the size in pixels is increasing linearly. Authors should clarify the discrepancy and depict the stimulus accurately in this figure.

Figure 5: Authors should indicate on the figure whether these differences are statistically significant.

Authors conducted 15 stimulus presentations for each crab. I suggest that they include the data for each individual crab's probability of hiding and the number of times it took for each crab to hide in the data.

Line 45, 56: Reference to Yilmaz and Meister (Curr Biol. 2013 Oct 21;23(20):2011-5. doi: 10.1016/j.cub.2013.08.015. Epub 2013 Oct 10.) would be appropriate as it is a more recent study than the ones the authors cited.

Experimental design

No comment

Validity of the findings

The authors claim in the abstract that the quicker the stimulus expanded the slower the rate of habituation, however in the results they say (line 223) “because large number of subjects failed to habituate, trials to habituation was not analyzed as a measure of reaction.” I suggest taking this out of the abstract as a main finding.

Additional comments

The authors presented the results using clear language. I suggest the paper will be ready for publication after the modifications stated above.

Reviewer 2 ·

Basic reporting

The paper is well-written and clear, and appropriate literature is cited. However, the organization of the Methods and Results sections are confusing and seem to overlap, and there seem to be multiple Results and Methods headings. I suggest that the authors have a methods section where they describe the experimental setup, animals, etc., and then a Results section where they describe all the results.

Experimental design

I'm curious as to why the first experiment used a star stimulus, and the second used naturalistic stimuli? That makes it difficult to ask whether the naturalistic stimulus was more effective. Did all the crabs escape in response to the hawk and crab images? If it's not possible to do additional experiments, I would not reject for this, but the data should be included if available.

Validity of the findings

It is not clear from figure 2 whether there was any statistically significant difference in the hide latency for the different expansion rates.

---

## Round 0.2 · accepted · Accept

· Academic Editor

Accept

I believe this paper will make a good contribution to the field. Thank you for your contribution to PeerJ.